# GENERATIVE MODELING FOR RNA SPLICING CODE PREDICTIONS AND DESIGN

## ABSTRACT

Alternative splicing (AS) of pre-mRNA splicing is a highly regulated process with diverse phenotypic effects ranging from changes in AS across tissues to numerous diseases. The ability to predict or manipulate AS has therefore been a long time goal in the RNA field with applications ranging from identifying novel regulatory mechanisms to designing therapeutic targets. Here we take advantage of generative model architectures to address both the prediction and design of RNA splicing condition-specific outcome. First, we construct a predictive model, TrASPr, which combines multiple transformers along with side information to predict splicing in a tissue specific manner. Then, we exploit TrASPr as on Oracle to produce labeled data for a Bayesian Optimization (BO) algorithm with a custom loss function for RNA splicing outcome design. We demonstrate TrASPr significantly outperforms recently published models and that it can identify relevant regulatory features which are also captured by the BO generative process.

## 1 INTRODUCTION

Alternative splicing (AS) occurs when multiple unique mRNA isoforms are produced from a single gene, by including or excluding different pre-mRNA exonic or intronic segments. AS greatly increases transcriptome complexity such that a single gene can encode many mRNA isoforms, each of which include a different subset of pre-mRNA segments. Over 90% of human genes undergo AS, with a conservative estimate that at least 35% of human genes switch their dominant isoform across 16 adult tissues Pan et al. (2008); Wang et al. (2008); González-Porta et al. (2013). Changes in the produced isoforms can have significant phenotypic effects: Defects in splicing have been associated with numerous diseases Singh & Cooper (2012) while at the molecular level, AS has been shown to change protein function by, for example, removing a nuclear localization signal, or affecting a binding domain of the encoded protein Smith & Valcárcel (2000); Licatalosi & Darnell (2010).

Following the discovery of RNA splicing in 1977 Berget et al. (1977); Chow et al. (1977), decades of work has identified hundreds of RNA Binding Proteins (RBPs) that regulate splicing outcome. These RBPs have been shown to bind exons and proximal introns, typically up to a few hundred bases away from proximal exons, to regulate splicing in a condition specific manner Fu & Ares Jr (2014a). See Appendix Figure 7d for an illustration. Consequently, a long-term goal for the RNA community has been to construct a predictive 'splicing code' that will be able, given a genomic sequence and cellular condition, to predict the splicing outcome Wang & Burge (2008).

Splicing outcomes are typically measured as the ratio of isoforms that include or exclude a specific RNA segment, referred to as 'percent spliced in' (PSI, or $\Psi \in [0, 1]$). The most common type of AS in human, which is the focus of this work, is cassette exons (see Appendix Figure 7 for an illustrative example). Similarly, for a cassette exon $e$ changes in splicing between two cellular conditions $c, c'$ (e.g., two cell types or w/wo genetic mutation) are expressed as dPSI $\Delta \Psi_{e,c,c'} \in [-1, 1]$. In this work, we consider two tasks related to splicing: Quantitative condition specific splicing prediction (i.e. $\Psi_{e,c}$, $\Delta \Psi_{e,c,c'}$ ), and splicing sequence design. The latter can be defined as generating a genomic sequence $S$ with a specific splicing profile across a variety of conditions $\{c\}$ to create for example a cassette exon $e$ which is highly included in brain but lowly included in other tissues. Of note, the genomic sequence $S$ may include intronic regions where regulatory elements reside to affect exon $e$ splicing, and it can be based on an existing exon $e$ that we want to 'enhance' or 'fix' with some genetic editing. The allowed editing 'budget' can be specified by the user to fit some biomedical

task (*e.g.*, CRISPR editing, ASO targeting), as no more than $N$ edit locations with no more than $M$ total base changes.

In terms of related work, the first splicing code models used manually curated regulatory featured derived from the literature Barash et al. (2010); Xiong et al. (2011; 2015). More high-throughput PSI quantification saw a shift from these earlier models based on boosted decision trees and Bayesian neural networks to deep neural networks Bretschneider et al. (2018); Cheng et al. (2021); Zeng & Li (2022). The best performance on tissue specific splicing prediction was achieved in Jha et al. (2017) using a similar set of predefined regulatory features that were first condensed using an Auto-Encoder, then combined in a MLP. Another line of works aimed to learn a code directly from the genomic sequence using a variety of architectures. MT-Splice for example used a CNN based architecture with 64 length-9 filters while the more recent Pangolin Zeng & Li (2022) employed a ResNet architecture originally introduced in the SpliceAI model for detecting cryptic splice site Jaganathan et al. (2019). Both MT-Splice and Pangolin focused on predicting mutations that affect splicing outcome and reported moderate accuracy for tissue-specific splicing prediction. Other works focused on predicting isoform usage given RBP expression levels Chan et al. (2021) and are therefore less relevant here.

The condition specific RNA splicing design task, formulated as a constraint optimization problem, is new and therefore lacks previous methods to directly compare to. In terms of related molecule design work, there has been much work on deep learning for protein design (c.f. Bepler & Berger (2021) for a review). Specifically, Bayesian Optimization approaches, as we introduce here for the splicing design, have been employed successfully for protein design (*e.g.*, . Maus et al. (2022b; 2023)). For RNA design tasks, previous work focused on the design of untranslated regions (UTR) in mRNA vaccines for optimal expression Castillo-Hair & Seelig (2022); Leppek et al. (2022) or the design of alternative polyadenylation Bogard et al. (2019). Algorithms employed in those works involved genetic algorithms for 5' UTR design Sample et al. (2019) and Deep Exploration Networks (DEN) Linder et al. (2019). The latter is the closest to the work presented here as it too uses a generative model. However, that work is quite different in terms of the model, the task and the data. The DEN model involves a FFN VAE which generates genomic sequence as a long position weight matrix (PWM) that is later collapsed to a sequence. For predictions, it uses either a CNN or a linear model of k-mer counts as in Rosenberg et al. (2015). The DEN task is to predict alternative 5' splice site selection and data used for it is a pool of $\sim$13,000 synthetic short sequences tested in cell lines.

The schematics in Appendix Figure 7 are meant to help orient readers less familiar with RNA splicing and also highlight some of the computational challenges involved in the tasks we address here. Of note, genomic regions that contain cassette exons can vary greatly in length from few hundred bases to many kilo-bases (Kb). Thus, applying Transformer based models 'out of the box' as was done in DNABERT Ji et al. (2021) for other tasks is challenging as these models span much shorter sequences. Indeed, even large CNN models such as SpliceAI and Pangolin only span 10Kb, which is less than the region spanned by 24 percent of the cassette exons we used here. Furthermore, such regions can involve multiple exons/junctions, making condition specific prediction of PSI/dPSI more challenging. In addition, the RBP binding sites are not well defined (low information content) and can either enhance or represses exon inclusion in a position dependent manner Fu & Ares Jr (2014b).

We address the above challenges by developing TrASPr, a new multi-Transformer based splicing code model. Rather than trying to apply generic 'splicing agnostic model' we first pre-train a Transformer on splice sites recognition. Then, we center a Transformer around each splice site region involved in a cassette exon (4 total), add additional genomic features to account for those (e.g intron and exon length, conservation), and combine the learned representation from each Transformer through several joint MLP layers. This combined model, TrASPr, is then trained on genome wide scans for cassette exons in the human and the mouse genome. We first test TrASPr on RNA splicing data from both mouse and human tissues, demonstrating it achieves state-of-the-art prediction accuracy compared to current models and alternative architectures that we assess via ablation studies. Then we show TrASPr detects condition specific regulatory elements using ENCODE data involving three RBP Knockdown (KD) in two human cell lines, and data for tissue-specific regulatory elements from a mini-gene reporter assay.

For RNA splicing design, we approach the optimization problem as described above by employing a deep generative model such that Bayesian Optimization (BO) techniques can be utilized for it. Specifically, our BO algorithm for splicing (BOS), uses TrASPr as an Oracle to optimize a VAE under sequence and splicing outcome constraints. We compare BOS to a few baselines, demonstrating

BOS can more effectively mutate a given sequence under a limited number of mutations to achieve a predefined tissue specific splicing outcome. Finally, we conclude with a discussion of applications, limitations, and future directions.

## 2 BACKGROUND

**Cassette exons detection and quantification:** Appendix Figure 7b illustrates the process of detection and quantification of cassette exons that serve as training data for splicing code models. Splicing quantification nowadays is mostly derived from Illumina RNA sequencing reads. Each experiments includes millions of short ( $\sim$100bp long) reads that are mapped back to the genome or transcriptome using dedicated tools (*e.g.*, STAR Dobin et al. (2013)). Dedicated splicing analysis algorithms are then used to first detect the AS events, typically from reads spanning across RNA segments (blue and red reads in Appendix Figure 7b cartoon), then quantify those in terms of $\Psi$ or $\Delta\Psi$ as described above. Here we applied the commonly used MAJIQ algorithm Vaquero-Garcia et al. (2016; 2023) to detect and quantify cassette exons from RNA-Seq data across different tissue types. See Appendix for more details regarding MAJIQ's usage and pipeline.

**Notation:** We measure splicing across $c \in [1, \ldots, C]$ conditions for events $e \in [1, \ldots, E]$. Each AS event $e$ has a sequence $S_e$ comprised of 4 different regions, each centered around the respective splice site $S_e = \{S_e^1, S_e^2, S_e^3, S_e^4, \}$. Similarly each event has a set of features associated with it such as exon length, conservation etc. denoted $F_e$. Splicing quantification for event $e$ in condition $c$ is denoted $\Psi_{e,c} \in [0, 1]$ and differential splicing as $\Delta\Psi_{e,c,c'} \in [-1, 1]$ accordingly. However we frequently drop the event $e$ or condition $c$ index for brevity.

**Pre-train and tissue specific splicing data:** To pretrain the basic BERT RNA model, we first extract 1.5 million sequences around splice sites from the GENCODE human pre-mRNA transcripts database. Each sequence was cut to be 400 bases long and centered around the splice site. These sequences are then converted into 6-mers tokens and fed as input to the BERT model.

For training and testing on tissue-specific cassette exons inclusion we use two main datasets. The first is from the mouse genome project (MGP)Keane et al. (2011) and involves six mouse tissues (Heart, Spleen, Thymus, Lung, Liver and Hippocampus) with 4-6 replicates each. We also used the same train/test data split used in Jha et al. (2017) so that the results can be compared directly to their model. The second dataset is GTExConsortium (2020) from which we select six representative tissues/conditions: Heart (Atrial Appendage), Brain (Cerebellum), Lung, Liver, Spleen, and EBV transformed lymphocytes. Note that some conditions are shared between the datasets. This ensures that our model sees sequences from different species but similar tissues. These datasets were all processed uniformly using MAJIQ as described above to detect cassette events with high-confidence quantification for their $\Psi_{e,c}, \Delta\Psi e, c, c'$. In total, we collect E=11346 and E=18278 events from the MGP and GTEx datasets.

The high number of cassette events in our data is partially due to the fact the cassette exons extracted from MAJIQ's splice graphs may be overlapping (*e.g.*, different splice sites used to define the skipped exon). This may be useful for training on diverse exon/intron definitions but care must be taken to avoid information leakage to the test data. We handle this issue by first hiding two chromosomes (8, 14 for GTEx and 4, 11 for MGP) for testing, then discard test exons that are too similar to training exons to avoid leakage due to paralogs or orthologs. We use two levels of filters for those: 'Standard', which are the same as those used in Jha et al. (2017) to control for overlapping events or close paralogs when using a single species, and 'strict' filters to control for more distant similarity across species (orthologs) as we used both mouse and human data here (See Appendix for more details).

**Mutations and knockdown data** To evaluate the capability of TrASPr and BOS to predict or suggest mutations, we curated two other sets of experimental data. The first one is the ENCODE dataset for RBP Knockdown (KD) in human cell lines Hitz et al. (2023). We focused on three well studied RBPs (TIA1, PTBP1, QKI) for which there is relatively better sequence motif definitions (to get a better estimate of binding sites locations) and better binding assays (eCLIP) which indicate regions where these RBPs were found to bind RNA *in vivo*. Several steps to correct batch effects Slaff et al. (2021) and filter for higher quality putative targets resulted in a list of 59 putative RBP regulatory targets (See Appendix for more details). We then 'removed' the effect of these RBPs on the set of AS targets by randomly mutating their identified binding motifs. We repeated this process 5 times with

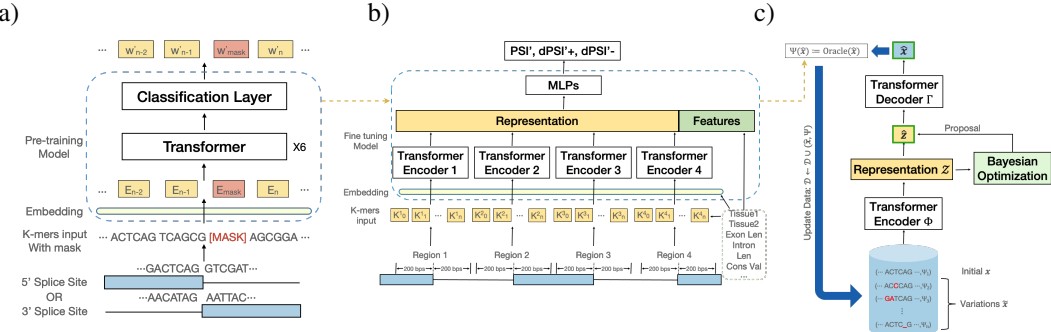

Figure 1: Pipeline and structures of our model. a) Pre-training stage. b) Fine-tuning TrASPr. c) BOS structure and flow.

different random mutations and the prediction results where then averaged and compared to the wild type (WT) sequence prediction. These *in-silico* predictions of RBPs effects where then compared to those observed in the actual ENCODE KD experiment. Finally, we also included experiments from a mini-gene reporter assay where the effect of mutating several regions upstream of exon 16 of the mouse Daam1 gene where tested Barash et al. (2010).

## 3 METHODS

Our method, depicted in Fig. 1, involves three main components: An elaborate data processing pipeline discussed above, a transformer based splicing prediction model (TrASPr), and a Bayesian Optimization algorithm (BOS) to design RNA with desired properties. We now turn to describe the two modeling components in order.

### 3.1 TRASPR

#### 3.1.1 PRE-TRAINING RNA SPLICE SITE BERT MODEL

The foundation model for TrASPr is a 6 layer BERT model which is pretrained on human RNA splice sites (Fig. 1b). Following the pretraining step, as in Ji et al. (2021), TrASPr takes an RNA sequence converted to 6-mer tokens as input, but instead of using the BERT default max length, we feed the model with 400 bases long sequences where the splice site (either 5' or 3' splice site, as shown in the cartoon) is in the center.

For pre-training, we follow BERT in randomly choosing 15% of tokens, but additionally mask the surrounding k tokens for each one to account for our overlapping 6-mer tokenization. We used standard masked autoencoding training, calculating the loss from the original 15% of tokens that were masked. We pretrain for 110k steps with a batch size of 40. The learning rate was set to 4e-4 and we used a linear scheduler with 10k warm-up steps.

#### 3.1.2 THE TRASPR MODEL AND FINE-TUNING

The structure of TrASPr is depicted in Fig 1c. For any given AS event $e$, the input to TrASPr is a sequence composed of four sequences $S_e = \{S_e^i\}_{i=1}^4$ such that each $S_e^i$ covers the exonic and intronic regions surrounding one of the four splice sites involved in the exon skipping AS event $e$. Each $S_e^i$ is fed through a matching pre-trained transformer $T^i$, which also accepts additional event features $F_e = \{F_{e,i}\}$ (see below). The latent space representation from each transformer $T^i$, captured by their respective CLS tokens, are concatenated together along with the feature set $F_e$ and fed into a 2 hidden layer MLP with width 3080 and 768.

**Event features.** The additional feature set $F_e$ includes the exon and intron length information as binned tokens, as well as the tissue type. We additionally include conservation values generated based on PhastCons scoreSiepel et al. (2005) for each k-mer in the sequence. Exons generally have significantly higher conservation values, as these reflect selection pressure due to non splicing related function (coding for proteins). We therefore used the mean of all conservation scores to fill the exon regions but kept the original scores for the introns.

**Supervision.** We follow Jha et al. (2017) and define targets based on measuring both splicing outcomes and *changes* in splicing outcome for an event $e$ in two different conditions $c, c'$. Specifically, the target variables included:

$$T_{\Psi_{e,c}} = E[\Psi_{e,c}], \; T_{\Delta\Psi_{+_{c,c'}}} = |\max(\epsilon, E[\Delta\Psi_{c,c'}])|, \; T_{\Delta\Psi_{-_{c,c'}}} = |\min(\epsilon, E[\Delta\Psi_{c,c'}])|$$

Here $E[\Psi_{e,c}], E[\Delta\Psi_{c,c'}]$ represent the posterior expected values for PSI and dPSI as estimated by MAJIQ from the RNA-Seq experiments Vaquero-Garcia et al. (2016). The $T_{\Delta\Psi_{+_{c,c'}}}$ target captures events with increased inclusion level between tissue c and c' while $T_{\Delta\Psi_{-_{c,c'}}}$ captures events with increased exclusion, forcing the model to focus its attention on those. To avoid gradient issue, we use random small number between 0.001 and 0.002 as $\epsilon$. For all of those target variables we use the cross-entropy loss function which performed better than regression. In the fine-tuning step, we train the model with 2e-5 learning rate and batch size of 32 for 10 epochs.

### 3.2 SEQUENCE DESIGN FOR SPLICING OUTCOMES.

Beyond supervised learning, we also demonstrate that TrASPr can be leveraged to solve sequence design problems. Given a sequence $S_e = (s_1, ..., s_n)$, TrASPr measures the probability that the splice site in the center of $S_e$ is included in some tissue $c$, $\Psi_c(S_e)$. This value can directly be used as the basis for optimization problems, where we seek to design new sequences $\tilde{S}_e$ that differ from $S_e$ only slightly, but exhibit altered splicing outcomes. Formally, we define these optimization problems:

$$\underset{\tilde{S}_e}{\arg\min} \, \Psi_c(\tilde{S}_e) \text{ s.t. } \text{lev}(\tilde{S}_e, S_e) \leq \tau \text{ or } \underset{\tilde{S}_e}{\arg\max} \, \Psi_c(\tilde{S}_e) \text{ s.t. } \text{lev}(\tilde{S}_e, S_e) \leq \tau \quad (1)$$

Here, $\text{lev}(\tilde{S}_e, S_e)$ denotes the Levenshtein distance between $\tilde{S}_e$ and $S_e$. Solving the minimization problem is equivalent to finding a small perturbation (up to edit distance $\tau$) of $S_e$ that *reduces* inclusion in the target tissue $c$ by as much as possible. The maximization problem corresponds to *increasing* inclusion. In practice, we add additional constraints that $\forall c' \neq c$, $\Psi_{c'}(\tilde{S}_e)$ cannot be reduced below 0.05. This additional constraint prevents an optimization routine from destroying splicing to such an extent that all inclusion levels are driven to zero.

To solve these optimization problems, we adapt recent work in latent space Bayesian optimization (LSBO) for black-box optimization problems over structured and discrete inputs Maus et al. (2022a); Stanton et al. (2022); Gligorijević et al. (2021); Moss et al. (2020); Winter et al. (2019); Sanchez-Lengeling & Aspuru-Guzik (2018); Gómez-Bombarelli et al. (2018); Griffiths & Hernández-Lobato (2020); Grosnit et al. (2021). LSBO solves structured optimization problems using two primary components: (1) a deep autoencoder (VAE) model, and (2) a Bayesian optimization routine.

**Variational autoencoders for LSBO.** In LSBO, we train a DAE that assists in reducing the discrete optimization problem over sequences $\mathcal{S}$ to a continuous optimization problem over the *latent space* of the VAE, $\mathcal{Z} \subset \mathbb{R}^d$. Leveraging the same data used to train TrASPr, we train a 6 layer Transformer encoder $\Phi : \mathcal{S} \to \mathcal{P}(\mathcal{Z})$ and 6 layer Transformer *decoder* $\Gamma : \mathcal{Z} \to \mathcal{P}(\mathcal{S})$ Vaswani et al. (2017). The encoder $\Phi(S_e)$ maps sequences $S_e$ onto a distribution over real-valued, continuous latent vectors $\mathbf{z}$. The decoder $\Gamma(\mathbf{z})$ (probabilistically) reverses this process. The parameters of $\Phi$ and $\Gamma$ are trained so that roughly we have $\Gamma(\Phi(S_e)) \approx S_e$. Because we only care ultimately about the output sequence $\tilde{S}_e$, here we abuse notation and denote the most probable sequence output from the decoder as $\Gamma(\mathbf{z})$. For optimization, the advantage the VAE provides is the ability to optimize over *latent vectors* $\mathbf{z}$ rather than directly over sequences $S_e$. This is because, for any $\mathbf{z}$ proposed by an optimization algorithm, we can evaluate $\Psi_c(\Gamma(\mathbf{z}))$. We therefore search for a $\tilde{\mathbf{z}}$ such that $\tilde{S}_e := \Gamma(\tilde{\mathbf{z}})$ is a high quality solution to the optimization problem.

**Bayesian optimization.** With the optimization problems in Equation 1 reduced to continuous problems over $\tilde{\mathbf{z}} \in \mathcal{Z}$, we can now apply standard continuous black-box optimization algorithms. Bayesian optimization Garnett (2023) is among the most well studied of these approaches in the machine learning literature. In iteration $n$ of Bayesian optimization, we have a dataset $\mathcal{D}_n = \{(\mathbf{z}_i, y_i)\}_{i=1}^n$ for which $y_i = \Psi_c(\Gamma(\mathbf{z}_i))$ is the known objective value. We train a surrogate model of the objective function using this data–most commonly a Gaussian process Rasmussen (2003)–and use this surrogate to inform a policy–commonly called an *acquisition function*–that determines what latent vectors $\mathbf{z}_{n+1}$ to consider next. In this paper, we use LOL-BO Maus et al. (2022a) as our base, off-the-shelf LS-BO algorithm. To accommodate the constraints in Equation 1, we modify

LOL-BO to utilize SCBO Eriksson & Poloczek (2021) rather than TuRBO Eriksson et al. (2019) as the underlying optimization routine. As with the objective, the Levenshtein constraint is evaluated on decoded latent vectors: $\text{lev}_{\mathcal{Z}}(\mathbf{z}, \mathbf{z}') = \text{lev}(\Gamma(\mathbf{z}), \Gamma(\mathbf{z}'))$.

## 4 RESULTS

In this section, we compare TrASPr with state-of-art methods on predicting condition specific splicing changes, assess its ability to predict the effect of changes in *trans* (RBP KD) or *cis* (mutations in a mini-gene reporter assay) using *in-silico*, then assess the ability of our proposed generative algorithm BOS to propose sensible sequences for a user defined splicing outcome.

### 4.1 PREDICTING EXON INCLUSION LEVELS ACROSS TISSUES

We first evaluate TrASPr on the task of predicting $\Psi$ using human GTEx data, comparing to Pangolin. Pangolin uses the SpliceAI model architecture Jaganathan et al. (2019) and was originally trained on data from four species, each with four tissues. Pangolin model is unable to define specific splicing events such as cassette exons. Instead it uses a 10Kb sequence window and predicts a 'splice usage' value for the position in the center, constructing a separate model for each tissue. To make Pangolin comparable, we feed the 3' and 5' splice site of each alternative exon $e$, then calculate the average of these two. Performance was evaluated on shared tissues and test chromosomes as used in Zeng & Li (2022). Our model achieved significantly higher Pearson correlation for PSI prediction (0.81 vs 0.17 see Fig 2), even though the training set is smaller due to only using overlapping tissues. Taking a closer look both models work well on most of low PSI cases. However, Pangolin performance suffered on some high inclusion cases, assigning low inclusion values. This result might be because of condition specific regulation, because the relevant sequence context is outside the 10kb fixed window used by Pangolin, or because other splicing signals in that window 'confused' the model with respect to quantifying the inclusion of the cassette exon. We note that as mentioned by the authors in Zeng & Li (2022), predictions for tissue specific splicing changes were not very accurate and we therefore not include them here.

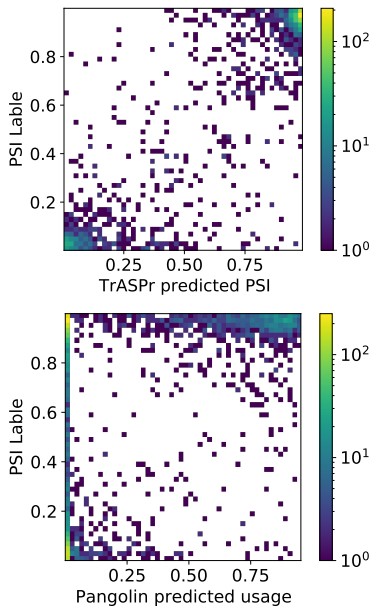

Figure 2: Comparison of PSI prediction results on GTEx dataset. Heatmaps show the distribution of prediction vs. RNA-Seq values for TrASPr (top, pearson 0.81) and Pangolin (bottom, pearson 0.173).

Next we turned to assess tissue specific differential splicing predictions. We compared TrASPr against a previous model that used the same target function but employed manually curated features parsed through an AutoEncoder and several layers of MLP (denoted 'AE+MLP feature model') Jha et al. (2017). This curated feature set was only available for the MGP dataset and so we assessed performance on this data using the same train and test set definitions as by the authors. Figure 3 and Table 1 show TrASPr significantly outperformed the AE+MLP model in identifying both differentially included and differentially excluded events, especially in terms of AUPRC (every pair of tissues is a point in the scatter plot with blue crosses and brown minuses each denoting evaluation on a set of differentially included or excluded events respectively). However, when we applied a more stringent filtering criteria on the test set TrASPr performance degraded while, surprisingly, AE+MLP performance improved. Inspecting the samples that were affected by the additional filtering, we found their labels generally correlated with those of somewhat similar training samples (hence giving TrASPr an advantage) but were not well predicted by the AE+MLP model which only used a predefined feature set.

### 4.2 ASSESSING TRASPR COMPONENTS

In this section we aimed to assess the contribution of various components of TrASPr through a series of ablation studies. First, we compared replacing our pre-trained BERT with the larger DNABERT

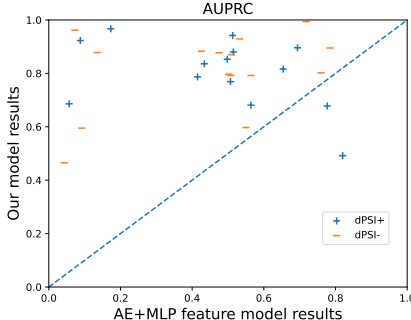

| | AE+MLP feature Model | | TrASPr | |
|---|---|---|---|---|
| Filter | Default | | | |
| AUPRC | 0.4861 | 0.4438 | 0.6079 | 0.6038 |
| Spearman | 0.5503 | | 0.6867 | |
| AUROC | 0.8712 | 0.8502 | 0.8895 | 0.8892 |
| Filter | Strict | | | |
| AUPRC | 0.5388 | 0.4874 | 0.5579 | 0.5176 |
| Spearman | 0.5962 | | 0.5917 | |
| AUROC | 0.8909 | 0.8766 | 0.8740 | 0.8695 |

Figure 3: Comparison of dPSI prediction results on MGP dataset

Table 1: Results on MGP dataset compared with AE+MLP feature model

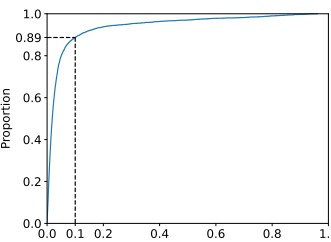 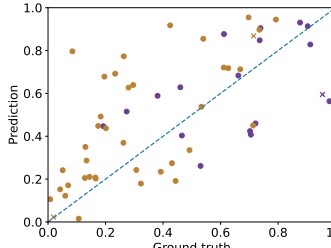 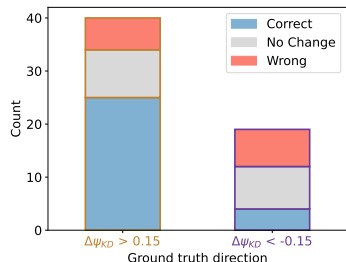

Figure 4: TrASPr prediction results on ENCODE dataset. Figures from left to right: (a) CDF of the difference between TrASPr predicted PSI and the ground truth on wile-type cases from GTEx+ENCODE test set. (b) TrASPr PSI prediction on wild-type AS events compared to RNA-Seq ground truth. Brown and purple indicates AS events whose inclusion are increased/decreased respectively upon RBP KD. (c) TrASPr dPSI direction prediction results for the events in (b). Blue, grey and red color bar means correct, no change, and wrong direction prediction respectively.

model trained on the entire human genome. One issue worth noting, was that the DNABERT model was highly sensitive to parameter settings and the pre-training failed with the parameters specified by the authors. We therefore did a parameter search and converted the max_grad_norm to 20 to avoid the vanishing gradient problem. Still TrASPr performed better even though the underlying BERT models were less complex. This result suggests that careful pre-training can be beneficial and that since coding sequence is only a small fraction of the human's DNA the DNABERT may be learning dependency structures which are less relevant for the task at hand.

Next, we assessed the model's performance with an ablation study to further understand the importance of each component of our model. As can be seen in the Table 2, when the model is trained from scratch, it has a hard time to extract more useful information from inputs and converges slower than the TrASPr. Therefore, even though it is similar to the full model on AUROC, it performs much worse by the other

Table 2: Ablation study. TrASPr - full model with pretrained transformers. noPre - Same structure and input as TrASPr but trained from scratch. noFeat - same train/pretrain as TrASPr but without extra features. wLSTM - model with a bidirectional LSTM instead of Transformer and without the extra features.

| | TrASPr | | noPre | | noFeat | | wLSTM | |
|---|---|---|---|---|---|---|---|---|
| AUPRC | 0.28 | 0.29 | 0.19 | 0.19 | 0.18 | 0.16 | 0.07 | 0.07 |
| Spearman | 0.33 | | 0.20 | | 0.30 | | 0.22 | |
| AUROC | 0.88 | 0.88 | 0.85 | 0.85 | 0.79 | 0.76 | 0.71 | 0.72 |

metrics. After removing the features, its performance significantly drops on prediction values but the ranking evaluation metrics does not decrease too much. To be noticed, the noFeat performance shows that simply applying a set of BERT models alone significantly drop the performance. Finally, if we use Bidirectional LSTM to replace transformers, the performance gets dramatically worse even compared to the no feature model. RNA splicing depends on complex regulatory elements around the splice sites. These results suggest an advantage of transformers models in extracting such information for the the splicing prediction task.

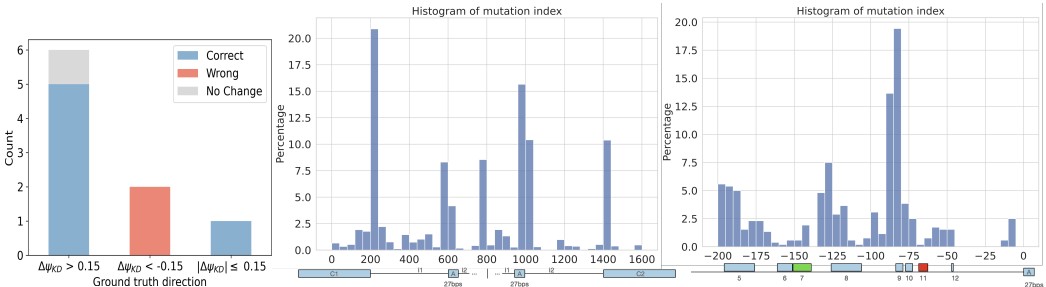

Figure 5: TrASPr dPSI prediction results on Daam1 gene. Figures from left to right: (a) TrASPr dPSI direction prediction results on 9 mutation regions of Daam1 gene. (b) Overall distribution of mutation hits generated by BOS. (c) Distribution of mutation hits among experiment regions.

### 4.3 PREDICTING THE EFFECT OF RBP KD AND MUTATIONS

We next turned to assess TrASPr ability to predict the effect of RBP KD and mutations. For this we first retrained the model using ENCODE data described in Section 2. First we assessed whether TrASPr is able to accurately predict exon inclusion in those new conditions. As shown in Figure 4a, TrASPr was able to predict $\Psi$ within 10% accuracy in almost 90% of the test cases, indicating the model was able to learn inclusion levels in those cell lines. Next, we focused on the set of putative RBP cassette exons targets shown in Figure 4b, where brown and purple represent events whose inclusion levels went up or down upon KD respectively. We find the WT $\Psi$ predictions for these correlated well with the experimental results (pearson's 0.65), and therefore continued with mutating the specific sequence motifs suspected to be the binding sites for the three RBPs of interest.

Before we could evaluate predictions for *in-silico* mutations we first needed to assess the significance of any given prediction. To achieve this, we randomly mutated sequences in the same set of exons, selected from the same distribution of distances as the original motifs (the distance can greatly affect the null distribution), but made sure non of these randomly chosen regions hit any of the 'real' motifs. We then used the 95 percentile of effects observed in this set as our threshold to call changes. The results of the *in-silico* mutagenesis experiment are summarized in Figure 4c. The left stacked bar shows cases whose PSI increased after RBP KD and the right bar shows decrease PSI cases. The correct(blue) and wrong(red) indicates if the predicted direction is the same as the label and no change(grey) means predicted dPSI was below the 95% cutoff described above. Overall, TrASPr performed well on most of the positive direction cases but predicted around half of negative direction cases as no change. The correlation coefficient for the dPSI effects was 0.34 with an associated p-value of 0.0192. The fraction of correctly called changes was over 50% with a p-value of 0.0001 (TNOM based test).

Finally, we assess TrASPr predictions for mutations introduced in a mini-gene reporter assay around a neural specific exon 16 in the mouse Daam1 gene. Similar to the ENCODE RBP analysis, we find TrASPr correctly predicts the effect of mutations in 7 out of 9 the cases (p-value 0.0012), as shown in Fig 5a. Here too, we find the model correctly predicts increased inclusion but the two mutations decreasing inclusion of exon 16 were not predicted correctly. We note these cases both involved region 11 (marked in red) which the model failed to capture.

### 4.4 ASSESSING BOS SEQUENCE GENERATION

We used TrASPr as an Oracle for our BOS algorithm to generate AS event sequences with edit distances from an original sequence of no more than $\tau = 30$. First, we asked BOS to increase the inclusion levels of lowly included cassette exons from Figure 4b. From the generated 214 sequences with increased inclusion (dPSI>0.2), our BOS algorithm significantly increased PSI(dPSI>0.5) for 46 of them. Most of the mutations were introduced around the relatively weak splice sites surrounding these AS events, which made biological sense. Scanning for the known motifs we found BOS also generated 15 cases where the known RBP regulatory motifs (TIA1, PTBP1 or QKI) were mutated to increase inclusion. When assessing BOS on the Daam1 exon 16 we again found many of the mutations increased inclusion by affecting the splice sites as expected (Figure 5b). However, zooming in on the upstream intron we found BOS frequently mutated the validated regulatory regions avoiding the region of small/little effect (green) and the area that caused decreased inclusion (red).

Next we assess the efficiency of BOS against two baselines in generating candidate sequences from a given input sequence and a user defined PSI constraint. Note that here we assume the Oracle is correct and only assess the ability to efficiently generate candidate sequences. The first baseline method randomly mutated 3, 6, 15 and 30-mers in different regions. We then calculated how many of these mutations actually changed the PSI by at least 0.2 based on the TrASPr oracle. In the end, the best random mutations setting (30-mers) successfully generated 177 out of 4392 sequences(4.03%). As a second baseline, we used the genetic algorithm (GA) from Sample et al. (2019), which was originally applied to design 5' UTR sequences. Generating the same amount of sequences the GA successfully generated 210 sequences (4.7%), while

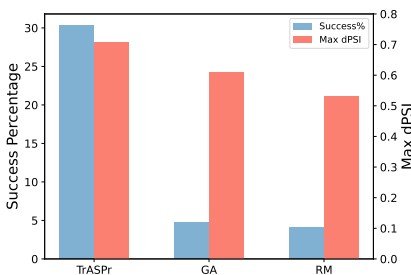

Figure 6: Comparison of RNA design performance. RM is the random mutation method and GA is the genetic algorithm method.

BOS generated 1331 successful sequences (30.3%) that matched the constraint of dPSI>0.2 (Figure 6). Furthermore, BOS also significantly outperformed the two baselines in terms of the best candidate sequence generated (0.71 maximum dPSI, compared to 0.61 0.53 for the GA and the random sampling algorithms). Overall, these preliminary results indicate that BOS is able to efficiently capture regulatory elements in a given sequence, including both splice site signals as well as deep intronic elements, then capitalize on those to generate sequences matching a given splicing target function.

## 5 DISCUSSION

In this study, we offer two main contributions. First, we propose a new tissue specific splicing code model, TrASPr. TrASPr leverages recent advances in LLMs utilizing Transformer based architecture. The architecture of TrASPr allows it on one hand to benefit from the Transformer attention mechanism while at the same time, by utilizing several Transformers each focused on a specific region, keep the model's attention on areas most relevant for splicing regulation without resorting to extremely large models. We demonstrated TrASPr was able to significantly improve performance in both PSI and dPSI predictions on several datasets compared to previous state of the art. These included CNN based models as well as models utilizing expert derived regulatory features that were fed into a DL model.

The second contribution in this study is in formulating the design of RNA sequences with specific splicing characteristics as a Bayesian Optimization problem. We then proposed the BOS algorithm, which uses TrASPr as an oracle, to solve this design problem with biologically plausible mutations. We showed BOS can effectively propose sequences that exhibit the desired splicing changes, mutating both core splicing signals and intronic regulatory elements.

It is important to keep in mind that the labels used for assessing the prediction tasks presented here are inherently noisy and limited in number. For example, RNA-Seq quantification is a noisy measurement, as are the RBP binding assays (eCLIP). The RBP regulatory motifs are crude as well. This means many targets might be missed while the changes upon RBP KD can be due to indirect affects (*e.g.*, another RBP affected by the KD) or other sequence motifs. Thus, the work presented here can be further improved, for example by incorporating high-throughput mutagensis experiments to tune it for the effect of disease causing mutations. Nonetheless, it is important to highlight applications of such tuned models in the near future. For example, recent work showed how splicing prediction algorithms can help detect effect of genetic variants to resolve undiagnosed rare disease cases Wagner et al. (2023). There, SpliceAI (which Pangolin tested here improved upon in Zeng & Li (2022)), appeared as a top performer. Another example is the identification of significant condition specific splicing variations in genes with poor RNA-Seq coverage. As for the RNA design task, BOS and similar algorithms could be leveraged for synthetic biology studies and for therapeutic design (*e.g.*, which sequence to target with ASO therapy or with prime editing).

**Code Availability:** Code and data will be made available upon publication

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

APPENDIX

## A  DETECTION AND QUANTIFICATION OF ALTERNATIVE SPLICING FROM RNA SEQUENCING

This section is meant to help orient readers less familiar with alternative splicing detection and quantification. Figure 7a shows several isoforms of the Daam1 gene in mouse. Notably, the genomic window containing the isoforms is over 160Kb long and the cassette exon region (red box) spans over 7.5Kb, illustrating the dimension of the problem. Detection and quantification of such AS events is mostly done nowadays with RNA sequencing where the RNA is first fragmented and then the fragments are sequenced using short reads (∼100b long) for one or two (paired) ends. These reads then need to be mapped back to the transcriptome or genome, as shown in Figure 7b by dedicated mappers such as STAR used here Dobin et al. (2013). Only a minority of those reads, shown in red and dark blue, are junction spanning reads that span across segments spliced together. Algorithms such as MAJIQ used in this work then use those junction spanning reads to quantify PSI or dPSI for specific segments or splice junctions as described in the main text. This $\Psi$ estimation is illustrated in Figure 7c as simple read ratios (1 red, 3 dark blue) though in practice a more sophisticated statistical model is requiredVaquero-Garcia et al. (2016; 2023). A splicing code task is then to take in the

genomic sequence in the region encompassing an AS event such as the cassette illustrated in Figure 7 and output a condition specific PSI or dPSI for changes of exon inclusion between two conditions $c, c'$. In practice, the outcome of splicing (PSI, dPSI) is the result of a complex multi-stage process illustrated in Figure 7d, where the spliceosome components recognize core signals (splice sites, branch point, poly-pyrimidine tract) while some of the hundreds of splicing regulatory factors, such as those of the hnRNP and SR families shown here, may bind in a condition specific manner exonic and intronic areas around the splice sites involved, enhancing or repressing the inclusion of a specific segment in the mature mRNA ( *cf* Fu & Ares Jr (2014b) for more detailed review).

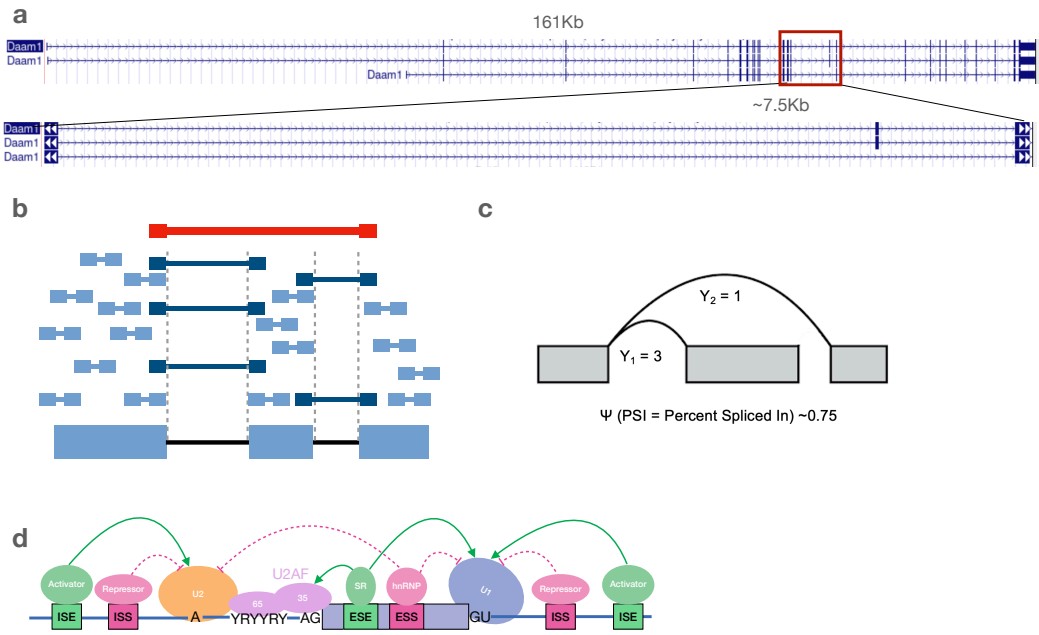

Figure 7: illustration of cassette exon alternative splicing. a) Isoforms of the Daam1 gene. b) Schematic of RNA sequencing and detection c) Corresponding PSI quantification for cassette exons. d) Schematic of components involved in RNA splicing and its regulation.

# B  MODEL DETAILS

## B.1  DATA FILTERING

To better evaluate the performance of our model, we removed the redundant sequences from the test set based on the similarity. The sequence similarity was assessed using BLAT Kent (2002) with filters for percent identify, difference in length, and the estimated similarity p-value. We consider two filter settings. First, we denote a set of 'Permissive' filters as used in , These settings included `maxLenDiff=5`, `minPval=0.0001` and `minIdentity=95`. Because we are using significantly more complex models, we introduce a second set of filters we denote 'Strict' with `maxLenDiff=100`, `minPval=0.001` and `minIdentity=80`. This accounts for short exons with high similarity but that diverge enough relative to their short length to not achieve a significant p-value.

## B.2  BATCH CORRECTION FOR ENCODE DATASET

ENCODE data involves two types of cell lines (K562, HepG2) in which various RBP were knocked down, followed by RNA-Seq experiments to measure the KD effect on the transcriptome. Since the ENCODE RNA-Seq data has been shown to exhibit strong batch effects we first performed batch correction using MOCCASIN Slaff et al. (2021). Here, we focused on three well studied RBPs (TIA1, PTBP1, QKI) for which there is relatively better sequence motif definitions (*i.e.*, which sequences

these RBP are likely to bind) and better experimental binding assays (eCLIP) which indicate regions where these RBPs were found to bind the RNA sequences. To assess whether the splicing code is learning direct regulation by these RBPs we searched for occurrences of these RBPs sequence motifs. Then we filtered those motif locations to be in AS events which had those in the intronic regions proximal to the alternative exon. We furthered filtered those for AS events that had eCLP binding peaks for those RBPs and that their inclusion level was indeed affected upon the RBP KD experiment ($|\Delta\Psi| > 0.15$)). This set of AS events served as putative targets of the above RBPs. We then 'removed' the effect of these RBPs on the set of AS targets by randomly mutating the identified binding motifs. We repeated this process 5 times with different random mutations and the prediction results where then averaged and compared to the wild type (WT) sequence prediction. These *in-silico* predictions of RBPs effects where then compared to those observed in the actual KD experiment. Finally, we also included experiments from a mini-gene reporter assay where the effect of mutating several regions upstream of exon 16 of the mouse Daam1 gene where tested Barash et al. (2010).

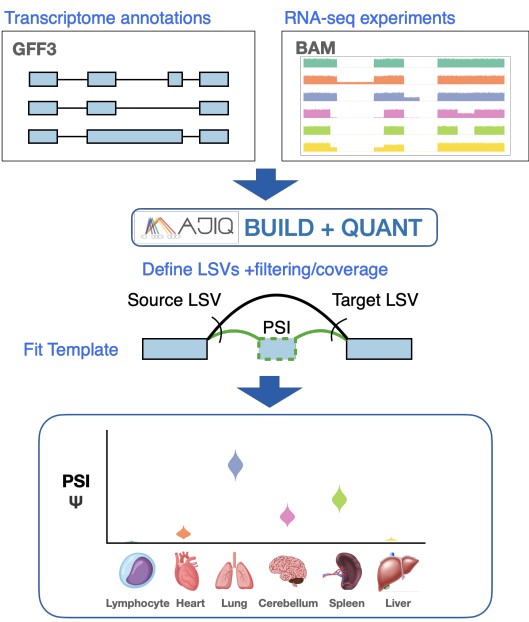

Figure 8: Cassette exons and PSI quantification pipeline.

### B.3 TRANSFORMER MODELING OF RNA SEQUENCE

In this work we adapt BERT Devlin et al. (2018) model to RNA sequences. BERT is a bi-directional transformer-based model, which learns contextual relations of tokens in a text Vaswani et al. (2017). The BERT model can be pre-trained on large unlabeled datasets of tokenized text using masked token prediction. Here we considered different tokenizing strategies of RNA sequences which are composed of 4 types of ribonucleotide bases ('A','C','G','U').

The importance of the choice of k-mer was tested in a preliminary experiment. We tested 4, 5 and 6-mer on the simpler task of just splice site prediction (same task as in the SpliceAI algorithm). We saw a monotonic increase in accuracy from the 4-mer (0.949) to the 6-mer (0.976) and therefore used 6-mer. Of note, the sequences typically associated with an RNA recognition motif (RRM) are ~5 bases long which is why we did not test very short k-mers. Inline with that, we saw 5 and 6-mer performed similarly with only a slight advantage for the 6-mer. Therefore, we settled on overlapping k-mers of length 6 such that the sequence "AUUGGCU" is represented by a string containing two tokens, `AUUGGC` and `UUGGCU`. During pre-training all k-mers that include a specific nucleotide are masked as was done in other BERT based genomic models (e.g. DNABERT Ji et al. (2021)). In addition to all possible 6-mer combinations of ribonucleotide bases, we include 5 special tokens to represent classification (`[CLS]`), padding (`[PAD]`), separation (`[SEP]`), mask (`[MASK]`) and unknown (`[UNK]`). Finally, we extend the vocabulary with additional tokens to capture additional features and information such as the tissue type, species and length tokens.

