# OpenReview forum: "Generative modeling for RNA splicing code predictions and design"
_ICLR.cc/2024/Conference — Submitted to ICLR 2024_

### Official Review · Reviewer_Bzy6 · 2023-10-29

**Soundness:** 2 fair
**Presentation:** 2 fair
**Contribution:** 3 good
**Rating:** 5
**Confidence:** 5

**Summary:**

Modeling alternative splicing (AS) and predicting the implications of _cis_-acting factors (e.g., proximal mutations) and _trans_-acting factors (e.g., RNA binding proteins; RBPs) have been studied for almost two decades. The splicing machinery and process are complicated and error-prone, which is why a significant fraction of diseases involve aberrant splicing. In this manuscript, the authors propose two major directions: using transformers for modeling alternative splicing (referred to as Traspr) and utilizing this model as an oracle to propose a Bayesian optimization (BO) method for designing sequences that achieve a desired alternative splicing level. They employ mutation and RBP knockdown data to demonstrate that their transformer-based model has learned the regulatory code of splicing. Finally, they illustrate, by example, how their Bayesian optimization method can alter alternative splicing levels.

**Strengths:**

While alternative splicing modeling has been extensively studied, systematic methods for designing/altering sequences are relatively new, and the authors' Bayesian optimization method appears promising.  RBP knockdown analysis is also interesting. The authors have also undertaken a significant amount of data processing and batch correction to facilitate model training and validation.

**Weaknesses:**

This manuscript attempts to address two major problems but, unfortunately, does not succeed in either case. Here are the main issues:

1. The authors focus on differential splicing and use splicing profiles across multiple tissues and species to train their model. However, they do not present any immediate benefits. For instance, they do not provide information on how much SpliceAI (Jaganathan et al., 2019) loses in performance, or how much Traspr gains, by having a universal (tissue-agnostic) model for splicing. In Section 4.1, the authors themselves acknowledge that "[...] predictions for tissue-specific splicing changes were not very accurate, and we, therefore, did not include them here."

2. The manuscript lacks convincing performance comparisons. It is challenging to believe that Traspr outperforms Pangolin (and subsequently SpliceAI) by such a significant margin (Spearman correlation 0.81 vs. 0.17). SpliceAI and other methods have undergone extensive testing by independent researchers. Furthermore, the approach for calculating cassette exon inclusion in Pangolin, as described in the manuscript, may not be optimal, as it ignores the upstream and downstream exon junctions. A more appropriate method can be found in the Methods section of the SpliceAI paper.

3. Traspr appears to excel at predicting constitutive and non-constitutive exons but performs poorly for exons that fall in between (the truly alternatively spliced exons). It might provide more insight if the tested exons were categorized accordingly (e.g., see Fig. 1C in the SpliceAI paper).

4. The manuscript lacks a comparison with the AE+MLP model, which is trained only on mouse data and is not considered state of the art.

5. The authors only examine Daam1 mutations published in (Barash et al., 2010), while numerous datasets have been published since then, e.g., in (Xiong et al. 2015) and (Jaganathan et al., 2019), among many others.

6. When evaluating the Bayesian optimization (BO) method for sequence design, other datasets could be considered, such as the SMA dataset in (Xiong et al. 2015).

Minor Comments:
- Not all figure panels are labeled, and some figure references are incorrect, e.g., Fig. 1b in subsection 3.1.1.
- Some tables have two values in some cells, and it is not clear what these values represent, e.g., in Tables 1 and 2.
- Fig. 1 is too small and challenging to read.

**Questions:**

I find the sequence design aspect of this manuscript interesting and promising. I have a couple of questions on that:

1. Is the Transformer model truly necessary here? Could the VAE be used to obtain representations and then be fed into the MLP? Essentially, could a single splice junction model be trained instead of a Transformer and a VAE?
2. On the same topic, it would be beneficial to understand how a model like SpliceAI, followed by an MLP, would perform as the oracle. I'm not convinced that the Transformer-based splice site detection (with a 400bp context) outperforms SpliceAI with a 10kb context.
3. Assuming that a tissue-specific splicing model works (which needs to be demonstrated), can the BO method be adapted to modify AS only in a specific tissue, such as the brain?

---

> ### Author Response · Authors · 2023-11-22
>
> We thank the reviewer for noting the promise in the BOS algorithm for designing sequences for RNA splicing and the significant amount of work that went into this manuscript. Due to the character limitation of each response, we will respond in multiple comments.
>
> ## Part 1:
>
> **Weaknesses:**
>
> >This manuscript attempts to address two major problems but, unfortunately, does not succeed in either case. Here are the main issues:
>
> The manuscript indeed addresses two major problems: (1) Tissue specific splicing predictions (2) Generative model for sequences that would exhibit desirable (tissue specific) splicing profile. We hold that we are indeed able to successfully address those problems in this work. We believe this rather strong language/statement by the reviewer is a compound result of (a) misunderstanding some of the results/analysis presented (b) an error we made in one of the results (“correlation 0.17”) which made it look suspicious to the reviewer, and consequently led them to doubt the entire manuscript (c ) additional evidence which the reviewer requests and that we believe we adequately address below. We break down our response to each specific point raised by the reviewer below. We apologize in advance for the length of this response - the comments and strong disbelief we felt reading those required a lot of work and analysis which we list below.
>
> >In Section 4.1, the authors themselves acknowledge that "[...] predictions for tissue-specific splicing changes were not very accurate, and we, therefore, did not include them here."
>
> This seems to be a misunderstanding - this comment in the manuscript was referring to a point made by the Pangolin authors about the fact Pangolin is not very accurate in predicting condition specific splicing changes - it is not a comment about TrASPr or about predicting tissue specific splicing changes in general. Regardless, we include those results now and as we clearly show below Pangolin does offer improved tissue specific splicing predictions over SpliceAI (after the above bug was fixed…) and TrASPr significantly outperformed both of those in this task.
>
> >Furthermore, the approach for calculating cassette exon inclusion in Pangolin, as described in the manuscript, may not be optimal, as it ignores the upstream and downstream exon junctions. A more appropriate method can be found in the Methods section of the SpliceAI paper.
>
> This seems to be a misunderstanding by the reviewer regarding how Pangolin computes inclusion. Pangolin uses SpliSER which does indeed take into account competing upstream/upstream junctions and computes a ratio between those going each splice site. SpliSER can be seen as a generalization of the direct approach the reviewer mentions in the Methods section of the SpliceAI paper, where multiple upstream/downstream splice junctions can be accounted for. Regardless, for the 3 GTEX tissues analysis described below for Pangolin we also retrained the model with MAJIQ’s PSI values (i.e. the same input used for TrASPr) and got no distinguishable differences on that dataset (changes of 0.01 and 0.02 in Pearson and Spearman correlations). Thus, we conclude that this is not an issue in our analysis.
>
> >The manuscript lacks convincing performance comparisons. It is challenging to believe that Traspr outperforms Pangolin (and subsequently SpliceAI) by such a significant margin (Pearson correlation 0.81 vs. 0.17). SpliceAI and other methods have undergone extensive testing by independent researchers.
>
> We believe this issue is likely at the heart of the reviewers general skeptical and negative assessment and therefore would like to address it first. A small correction to the Reviewer’s point is that the claimed 0.17 was Pearson (not Spearman). We wrote: “Pearson correlation for PSI prediction (0.81 vs 0.17 see Fig2)”.  We actually did exactly the same mistake ourselves which was Error1 by us: The scatter plot shown in Fig2 had Pearson correlation of 0.34. The 0.17 value was the Spearman value, with rank correlation suffering more from the bulge of points at the top of the scatter plot. This “bulge” leads us to the much more severe Error2 we made. This has to do with differences in input data processing between SpliceAI and Pangolin which messed up the Pangolin performance. We only caught this mistake after submission and we deeply apologize for not catching it before. The actual numbers are indeed much higher, and the scatter plot does look better (will be included in the final version, if given the opportunity). Again, we believe this error set the tone for much of the review and we deeply apologize for not catching it early. We hope both the correction and the many additional analyses detailed below will address the “ lacks convincing performance comparisons” view of the reviewer.
>
> (continued in the next part)

---

> ### Author Response · Authors · 2023-11-22
>
> ## Part2:
>
> >The authors focus on differential splicing and use splicing profiles across multiple tissues and species to train their model. However, they do not present any immediate benefits. For instance, they do not provide information on how much SpliceAI (Jaganathan et al., 2019) loses in performance, or how much Traspr gains, by having a universal (tissue-agnostic) model for splicing.
>
> Trying to address the above concern we see two potential arguments being made here by the Reviewer.
> The first potential argument being made here is that there is no point (“immediate benefits”) in modeling tissue specific splicing. For this, we would point to decades long research about the functional importance and overall abundance of tissue splicing, how it is tightly regulated and how changes in tissue specific splicing (or targeting it) are key in disease and for therapeutics. We are happy to make that point more clear in the revised version but do not believe this requires additional proof in the scope of this paper.
>
> The second argument, which seems more likely to follow from the Reviewer’s suggestion to compare to SpliceAI, is that even if predicting tissue specific splicing is desirable (see above point), a strong tissue agnostic algorithm such as SpliceAI may do just as well. We agree we should have better addressed this concern and therefore performed the tests described below. Before we describe those in detail below, it is important to note the following: If you take a random set of AS events (as was done here and in many other studies) across a few tissues, most events are NOT tissue specific. Thus, a good predictor of their non-tissue specific PSI should do well (making the point about SpliceAI comparison). However, being able to predict tissue-specific events is nonetheless an important task even if those cases are the smaller set (see point above). Now for the additional analysis and results:
>
> We ran the three algorithms in question (SpliceAI, Pangolin, TrASPr) on AS events from the test chromosomes as described in the paper, for the 3 tissues that matched those in the Pangolin paper. We term this dataset GTEX-3. The results of these tests were as follows (all results include two numbers: first Pearson, then the Spearman correlation)
>
>            |  TrASPr   | Pangolin  |  SpliceAI
>      GTX-3 | 0.91,0.81 |  0.8,0.67 |  0.69,0.6
>
> We find that Pangolin (note the numbers here reflect the bug fix described above), which uses the SpliceAI architecture but is trained with tissue information, indeed offers improvements over SpliceAI in PSI prediction in those tissues. TrASPr significantly outperforms both models. The improvement of TrASPr over Pangolin is also much more significant than Pangolin over SpliceAI. Also, note that as described above we also tried re training Pangolin with MAJIQ PSI values for these 3 tissues, with similar results. Overall, these results suggest that the change in model architecture is key to achieve this improved PSI prediction performance across the 3 tissues.
>
> While the above shows significant improvement by TrASPr, most events in the GTEX-3 test set do not change PSI significantly between a pair of conditions/Tissues. Specifically, in the GTEX-3 test set only ~25% of the tissue pairs comparison in this set show a change of dPSI >= 0.15 so performance on this is governed to a large extent by a good fit for PSI values more than tissue specific effects. To zoom in on those important subset of events we defined the test set GTEX-3C: The same test set as above but now focusing only on the ~¼ of pairwise cases that exhibit dPSI >= 0.15.
>
>            |  TrASPr   |  Pangolin  |  SpliceAI
>     GTX-3C | 0.75,0.73 |  0.59,0.55 |  0.41, 0.39
>
> Here we see more clearly the benefit of tissue specific modeling when events are indeed changing, at least to some extent (dPSI >= 0.15). Still, this test reflects PSI prediction on changing events and does not measure directly the ability to classify changes. We thus computed AUROC and AUPRC (since there is class imbalance) for classifying both events that are changing “Up” (dPSI+) between two tissues and those that are changing “Down” (dPSI-) between two tissues. Since this is the GTEX-3 dataset (i.e. the 3 tissues available in Pangolin) we have 3 pairs of tissues so the stats below are the average of those (first number is for dPSI+ second number is for dPSI-)
>
> **TrASPr:**
>
> AUPRC: [0.37 0.50]
>
> AUROC: [0.83 0.88]
>
> **Pangolin:**
>
> AUPRC: [0.15 0.23]
>
> AUROC: [0.60 0.68 ]
>
> We see a significant improvement with TrASPr. Note that by definition SpliceAI can not be applied here.
>
> (continued in the next part)

---

> ### Author Response · Authors · 2023-11-22
>
> ## Part3:
>
> Finally, to demonstrate that the above results are not tied to a specific dataset/species, we repeated this analysis with TrASPr over the mouse 6 tissues dataset (MS-6) where ~23% of the sample pairs across tissues showed dPSI >= 0.15. We repeated this for the subset of changing samples (MS-6C).
>
>          |  TrASPr
>     MS-6 | 0.79,0.45 |
>     MS-6C| 0.60,0.57 |
>
> While performance here is different (likely due to different quality in terms of #reads and #samples), the trends are the same. Taken together, we believe that we added a substantial amount of analysis/results that clearly support our assertion that (a) There is benefit in tissue specific splicing prediction (b) We indeed achieved SOTA results for this task.
>
> >Traspr appears to excel at predicting constitutive and non-constitutive exons but performs poorly for exons that fall in between (the truly alternatively spliced exons). It might provide more insight if the tested exons were categorized accordingly (e.g., see Fig. 1C in the SpliceAI paper).
>
> The above comment seems to reflect a fundamental misunderstanding of the analysis/data which is a clear indication we failed to explain it well. Alternatively, there is a lack of clarity regarding the terms “constitutive” and “non-constitutive” used above by the Reviewer. Either way, let us try to clarify this important point. First,  ALL the data presented in this work is for alternative cassette exons. Alternative exons are defined as those for which we have evidence of being alternative based on the annotation and/or read data as processed by the MAJIQ algorithm for the specified tissues. Conversely, “Constitutive” are those that appear as such in the annotation and don’t have read data supporting otherwise. Notably, these labels are by definition somewhat noisy and context specific: A constitutive exon may become alternative in some other context and an alternative one may have very little reads (or just annotation) supporting it as such while in practice it is very much “constitutive”. Distinguishing between alternative and constitutive exons (originally based on annotation alone) is considered a relatively easy task in the splicing field with AUC well over 90% in SVM papers dating back to 2005 (Dror et al Bioinformatics) and later in Splicing code papers as well (~97% AUC, Barash et al Genome Biology 2014). This is NOT the task we are handling here as all exons are “alternative” by the above definition. In fact, if we were to introduce such constitutive exons into our test set our performance would skyrocket (as discussed in the GB 2014 paper above). This is also not about scoring (cryptic/weak) splice sites vs ‘main’ ones and mutations that activate/decrease those (as in the SpliceAI paper). Consequently, referring to Fig 1C in the SpliceAI the reviewer pointed us to, we see a significant difference: The vast majority of inclusion rates in that figure (>85%) are 0.9-1.0. In contrast, the data in the GTEX-3 set we described above, based on alternative exons in those tissues, is distributed more evenly with ~50%  at the 0.9-1.0 range and the rest are roughly evenly distributed between the 0-0.1 and the 0.1-0.9 ranges.
>
> Another option is that the Reviewer made here a general comment that high inclusion and high exclusion are generally easier to predict. We would be the first to agree (see above stats from scoring constitutive vs. alternative exons) and there are of course biological reasons for that - easier to detect a crappy splice site or a very strong (aka constitutive) exon. Furthermore, “in between” cases as the Reviewer calls them are more likely to be those that are shifting in a tissue specific manner - these are exactly the ones we are trying to better predict here.
>
> In conclusion, predicting alternative vs. constitutive is a much easier task then predicting exact PSI for alternative only events (which is the case here), and predicting ‘in between’ cases specifically is harder as these are harder to detect and many times reflect tissue specific signals. However, we strongly disagree with the negative conclusion “performs poorly” - compared to what? For sure we could do batter, but we believe we show strong evidence we actually perform better than previous algorithms on those in between values (see for example the GTEX-3C evaluations above when things by definition shift from 0 and 1 and classifying tissue specific changes).
>
> (continued in the next part)

---

> > ### Author Response · Authors · 2023-11-22
> >
> > ## Part4:
> >
> > >The manuscript lacks a comparison with the AE+MLP model, which is trained only on mouse data and is not considered state of the art.
> >
> > We were uncertain what the specific concern raised here was. We included AE+MLP on the mouse data - the only dataset available for this model which requires pre-processed manually curated features (no human data for those features). Given the high performance of this model on tissue specific predictions we stand by our definition that it was indeed “state of the art” for this specific prediction task - see comparison of it to TrASPr and comparison of TrASPr to SpliceAI and Pangolin above. Regardless, we now include tissue specific predictions comparisons to Pangolin as well.
> >
> > >The authors only examine Daam1 mutations published in (Barash et al., 2010), while numerous datasets have been published since then, e.g., in (Xiong et al. 2015) and (Jaganathan et al., 2019), among many others.
> >
> > The reviewer brings up an important point which seems that we failed to communicate clearly. Let us try to amend this here (and, if given the opportunity, in the revised version). The tasks in this paper were to (a) predict tissue specific splicing changes (b) build generative models to generate sequences with tissue specific splicing profiles. We stress the task was NOT to assess mutations - we emphasized this as a direction of future work in the discussion, well outside the scope of this work. The only reason to include the Daam1 results, and the ENCODE RBP analysis, was to show the model is capturing tissue specific regulatory signals. The mutations in Xiong et al 2015 and Jaganathan et al 2019 are NOT, to the best of our knowledge, for tissue specific splicing regulation and therefore we did not see a specific reason to include these (and similar datasets) in our results.
> >
> > With that said, the combination of high confidence (level 5) and skeptical tone have led us to believe that if we don’t do additional mutation analysis the reviewer will again suspect our work. Thus, we have worked hard to include a much more comprehensive mutagenesis dataset. We decided to focus our effort on a recent dataset from Cortés-López et al Nat Comm 2022. We choose this dataset because: (1) It focuses on splicing of CD19 exon 2 which has significant implications for immunotherapy treatment (2) Because of that, the authors included exon 1-2-3 in their mini gene construct, including the flanking introns (3) consequently it has >10K data points of random mutations (SNP/indels) combinations across both the exon and the flanking intron (no specific selection bias), > 4K mutation positions with 90% of the positions covered by at least 4 different mutation combinations (4) unlike some other studies the authors quantified PSI directly (and not some related measure) and the replicates had > 0.9 correlations. We call this dataset the CD19 data for short.
> >
> > After some data cleaning/filtering for consistent measurements etc. we first trained TrASPr on the cell line used in this study (NLM6) and then fine tuned it to predict the CD19 data. We evaluated TrASPr on this dataset using 3 different test set definitions:
> >
> > *RCV*: Random cross validation. This means the model may have seen a specific mutation, but not the specific combination of mutations in the test set.
> >
> > *SPF*: Single position filtering. Here the test data involves mutation which we have never seen before in any training case. Due to this constraint the data is more limited: Training set: 4628, Test set:1480
> >
> > *SPFWN*: Single position filtering with neighbors. Here we do not only hide the test positions, but also any position that ended up in a mutation combination with those positions. The nature of the dataset (mutation combination) is such that even when we do this for positions with least amount of shared mutations we still end up with Training set: 2300, Test set: 101.
> >
> > TrASPr results for these tests (PSI Pearson correlation) are:
> >
> > RCV: 0.93
> >
> > SPF: 0.91
> >
> > SPFWN: 0.85
> >
> > Using the same test sets on Pangolin and SpliceAI we get:
> >
> > **Splice AI:**
> >
> > SPF 0.7
> >
> > SPFWN: 0.75
> >
> > **Pangolin:**
> >
> > SPF 0.58
> >
> > SPFWN: 0.44
> >
> > (continued in the next part)

---

> > > ### Author Response · Authors · 2023-11-22
> > >
> > > ## Part 5:
> > >
> > > Note that both SpliceAI and Pangolin are used “as is” on this dataset just for illustrative purposes. SpliceAI does not offer condition specific training and the fluctuations between SPF and SPFWN in that case are likely due to the differences in test sets. In contrast, for Pangolin using “as is” meant averaging the predictions for the available tissues, which are likely very different from the NLM6 cell line and thus causing degraded performance compared to SpliceAI. It also seems likely re-training Pangolin on this cell line and mutation training set will significantly improve its performance. However, all of this is well outside the scope of this work and we never make any claims here that TrASPr is better at this task. We do believe that here is great potential for it, as we mention in the discussion, but the only reason to include this detailed analysis was simply to avoid the Reviewer suspicion we are hiding/misrepresenting our work.
> > >
> > > >When evaluating the Bayesian optimization (BO) method for sequence design, other datasets could be considered, such as the SMA dataset in (Xiong et al. 2015).
> > >
> > > Again, that data is not tissue specific and our main analysis is about constraining BOS to tissue specific signals. We agree additional assessment of that sort for BOS would be nice but we simply were not able to find a good/easy to process dataset for that and definitely not able to complete any additional analysis with the week long of rebuttal time.
> > >
> > > >Is the Transformer model truly necessary here? Could the VAE be used to obtain representations and then be fed into the MLP? Essentially, could a single splice junction model be trained instead of a Transformer and a VAE?
> > >
> > > We have not explored many alternative designs for BOS, we agree it would be interesting to explore alternative design choices in the future.
> > >
> > > >On the same topic, it would be beneficial to understand how a model like SpliceAI, followed by an MLP, would perform as the oracle. I'm not convinced that the Transformer-based splice site detection (with a 400bp context) outperforms SpliceAI with a 10kb context.
> > >
> > > We hope our detailed analyses described above points to the benefit of using TrASPr and not SpliceAI for designing sequences with a tissue specific splicing profile. The last comment here (“I'm not convinced that the Transformer-based splice site detection with a 400bp context outperforms SpliceAI with a 10kb context.”) is worth discussing though: The reviewer seems to mix here sheer attention region/model size with model choices for different tasks. This is important because this, again, may have led the reviewer to doubt our work. Let us revisit this point: SpliceAI was designed for a very different task - detecting splice sites as these relate to mutations. It has no prior knowledge about where those splice sites are and its approach is “splicing agnostic”: it scans large genomic windows of 10kb long trying to figure out if the middle position will be a splice site yes/no. This is an important and hard task for sure (e.g. disease diagnostics), and SpliceAI excels at it. TrASPr is designed for a very different task - predicing tissue specific splicing changes. Tissue specific splicing changes are hard to predict (see discussion and results above and in the paper) but TrASPr already “knows” where the splice sites are (based on annotation and/or RNASeq information) and we know from decades of research that most of the regulation occurs around those sites. Thus it can “afford” to have only 400bp windows of attention around those splice sites, and yet outperform SpliceAI in this task. Furthermore, as we write in the paper, 10kb is not even enough to cover many of the regions that involve cassette exons, which puts SpliceAI/Pangolin at a disadvantage for this task.
> > >
> > > >Assuming that a tissue-specific splicing model works (which needs to be demonstrated), can the BO method be adapted to modify AS only in a specific tissue, such as the brain?
> > >
> > > Yes! This is exactly the kind of constraint on BOS that we performed in the paper. Revising the original submission we saw the only description we included was “generating candidate sequences from a given input sequence and a user defined PSI constraint”. The description of that user constraint (for tissue specificity) was omitted. We will fix this in the revised version, given the opportunity.
> > >
> > >
> > >
> > > In conclusion, We hope the extensive responses we included to all reviewers will convince this reviewer the work does pass the acceptance threshold for a short ML conference paper and that the revised manuscript indeed demonstrate clearly that we addressed two major problems: (1) SOTA for tissue specific splicing predictions (2) Generative model for sequences that would exhibit desirable (tissue specific) splicing profile. We thank the reviewer for their detailed comments, time/effort on this, and their consideration.

---

> ### Comment · Reviewer_Bzy6 · 2023-11-22
> **Promising ideas that need a better focus on more polish**
>
> First, I need to thank the authors for the immense amount of time that they have put on answering reviewers' questions. And I do believe most of them should have been in the paper to begin with. For example, the focus on dPSI >= 0.15 might be a much more convincing experiment than what is already in the paper.
>
> Some of the explanations help me understand the paper better, but I feel my original comments still apply. The authors attempt to solve two hard computational biology problems, which results in suboptimal results for both. This might be better suited for a Comp Bio journal rather than this venue.
>
> To summarize my comments and reiterate my points, let's focus on Section 4.1 and Fig. 2. Here the goal is predict PSI in different  tissues and it is essentially the main benchmark.
> - What does each bin in the 2D histogram show? Are all (tissue, AS event) included in this plot? How does the tissue-stratified PSI prediction error look like?
> - Looking at Traspr results and if one excludes PSI < 0.1 and PSI > 0.9, correlations look much weaker, judging by the density of points at the small and large PSI values. Again, focusing on dSPI rather than PSI could be helpful here.
> - Tissue-specific splicing can also be switch-like: an exon is always included in one tissue and never included in other tissues. Showing dSPI instead of PSI can be helpful.
> - It looks peculiar that Pangolin does such as bad job here. It predicts an almost-zero PSI for hundreds if not thousands of events, that even basic algorithms don't. Is PSI zero in one tissue or in all tissues? I think the devil is in how splice site usage is translated into PSI.
>
> I will finish by saying that the BOS method looks super promising and I am confident with the right experimental setup it can be an impactful method.

---

> > ### Author Response · Authors · 2023-11-22
> > **Re: Promising ideas that need a better focus on more polish**
> >
> > First, we thank the reviewer for responding before the holiday hits - much appreciated!
> > We also thank them for pointing out the amount of work that went into the response and additional analysis.
> > As for the specific comments:
> > *"most of them should have been in the paper to begin with"*  - Yes, but hindsight is always 20-20. We did run this paper through others before, addressed everything that was suggested, and still not of that came up. Reviews make papers better, and your comments/feedback have pushed us strong to do just that so we again thank you for the time and feedback.
> >
> > *"I feel my original comments still apply"* - We are not sure what from the original comments still apply? We believe we have addressed all issues raised before and the ones below are additional questions we are happy to answer as well.
> >
> > **Section 4.1 Fig2:**
> >
> > Please note what the reviewer is referring to *is still the original submission* - it misses all the additional information and analysis we included in the response. Again, we would be happy to include those in an updated version given the opportunity but the reviewer can surely see why we are hesitant to update the manuscript (a major effort to organize things into a main and supp text/figures) if the reviewer maintains a negative view of our work.
> >
> > *"What does each bin in the 2D histogram show? Are all (tissue, AS event) included in this plot? How does the tissue-stratified PSI prediction error look like?"*
> >
> > There are no histograms in Fig2 and there is no subfigure 2D in the paper, so we are a bit confused. The question seems to refer to the scatter plot in Fig2? If so, **please note this figure as we wrote before has the bug in Pangolin output which we reported**.
> > To answer the question the points are not tissue stratified, we can add those in supp figures, but we do not expect to see a big difference between tissues. As we discussed in the response and the reviewer notes here as well, these plots are dominated by points close to 0 and 1 so no big difference in PSI scatter plots between tissues.
> >
> > *" if one excludes PSI < 0.1 and PSI > 0.9, correlations look much weaker, judging by the density of points at the small and large PSI values."*
> >
> > Of course. And we point it out in our response - see how correlation coefficients drop in our reported new results when only changing events are considered. That said, we believe the reviewer should consider the context of this problem. Yes, strong correlation is driven by the values around 0.1 and 0.9 and they are worse for "in between", but that is a known problem in PSI prediction and tissue specific changes predictions. Not only are we not alone in finding "in between" more challenging, we encourage the reviewer to revisit figures in high-profile papers claiming state of the art results to calibrate their expectation for this specific task. Specifically, look at:
> >
> > Figure 5A in Rosenberg et al Cell 2015
> >
> > Figure 6 A/B in MTSplice (another dedicated tissue specific splicing algorithm using DL) Cheng et al 2020
> >
> > Supp FigS4 in Pangolin (notice this is only in the supp figures) Zeng and Li GB 2022
> >
> > To iterate, we show by a multitude of metrics that we achieve SOTA in this task. So yes, we are far from perfect and we are happy to elaborate on this in a revised version, but we don't think this fact, taken in the context of this specific task and what has been achieved so far, should be held against us.
> >
> > *"Again, focusing on dSPI rather than PSI could be helpful here."*
> > There is value in showing both PSI and dPSI related statistics. We show PSI plots because (a) biologists care about those and how well you predict those (b) biologists care about the ability to predict tissue specific splicing changes - which is why we assessed AUROC, PRAUC and included correlation stats for changing events. We can also add correlation plots for dPSI in the supp figures if the reviewer believe this is necessary.
> >
> > *"Tissue-specific splicing can also be switch-like....Showing dSPI instead of PSI can be helpful."* - Please see above response.
> >
> > *"It looks peculiar that Pangolin does such as bad job here."*
> > Yes, this is exactly what we pointed out in our original response! This was a bug and we fixed it, see new stats we provided. We are unable to include an image in the response but it will of course be included if we are given the opportunity.
> >
> > *"This might be better suited for a Comp Bio journal rather than this venue"*
> >
> > Possibly, but we personally would want more analysis and results in a full journal paper. This paper aims to introduce the ML community to new and promising approaches for modeling tissue specific splicing and RNA sequence generation. We believe we achieved this goal and hope the reviewer can agree with us on this. If that is not the case we ask to please point out where we fail.
> >
> > Thank you again for the time, effort, and consideration of our work.

---

### Official Review · Reviewer_stAa · 2023-10-31

**Soundness:** 3 good
**Presentation:** 3 good
**Contribution:** 3 good
**Rating:** 6
**Confidence:** 4

**Summary:**

The authors propose a model for condition-specific alternative splicing quantification, improving on existing work for this task.  They also demonstrate the utility of the model for a synthetic biology problem of modifying a genomic sequence to modify the splicing pattern in a desired way.

**Strengths:**

- Prediction of condition-specific alternative splicing is an important biological problem, and the proposed model performs MUCH better than the recently published Pangolin model.

- The application of the model to the problem of modifying splicing patterns in a desired way is highly novel and interesting.  While deep learning models have been used for sequence design, the particular application is new.

- An ablation study demonstrates the value of the major aspects of the architecture, and contains interesting insight on the utility of DNABERT vs an RNA counterpart trained by the authors.

- Validation of model prediction using knockdown data.

**Weaknesses:**

- The ability to assess the validity of the mutated sequences is rather limited.  The authors were able to show that alternative methods for exploring sequence space to find mutated sequences are less effective at that.

- Although overall the manuscript is clear and easy to read, it contains many typos and grammatical errors - see below.

**Questions:**

- When it comes to in-silico mutations, I am not sure the authors' experiment shows all that much.  A better approach might be to see how well the model predicts splicing QTLs as was done in the Pangolin paper (and using integrated gradients works better than mutagenesis in our experience).


Minor comments:

"For all of those target variables we use the cross-entropy loss function which performed better than regression"
Do you mean better than a regression loss function?

"This result might be because of condition specific regulation, because the relevant sequence context is outside the 10kb fixed window used by Pangolin, or because other splicing signals in that window ‘confused’ the model with respect to quantifying the inclusion of the cassette exon. "
While the first reason seems plausible, the second is very generic and would apply to any model.

In section 4.3 it would be more friendly to the reader to spell out the first occurrence of KD since it's not a standard acronym.


typos/grammar:

curated regulatory featured --> features

constraint optimization problem --> constrained

6-mers tokens --> 6-mer tokens

and the prediction results where then averaged --> were

predictions of RBPs effects where --> RBP effects were

Daam1 gene where tested --> were tested

The foundation model for TrASPr is a 6 layer BERT model which is pretrained on human RNA splice sites (Fig. 1b).  --> 1a

The structure of TrASPr is depicted in Fig 1c. --> 1b

please improve the following sentence:
assess its ability to predict the effect of changes in trans (RBP KD) or cis (mutations in a mini-gene reporter assay) using in-silico

Figure 2 caption:
lable --> label

Pangolin model is unable --> The Pangolin model is unable

work well on most of low PSI cases --> work well on most low PSI cases.

advantage of transformers models --> transformer

 in extracting such information for the the splicing prediction task.

results on Daam1 gene --> results on the Daam1 gene

but made sure non of these randomly chosen regions hit --> none

---

> ### Author Response · Authors · 2023-11-22
>
> We thank the reviewer for the time and effort spent to review and comment on our work. We were very pleased to find such a nice summary of key points as points of strength in our work - Thank you!
>
> We focus below only on the main points raised that require fixing/clarifications.
>
> >The ability to assess the validity of the mutated sequences is rather limited. The authors were able to show that alternative methods for exploring sequence space to find mutated sequences are less effective at that.
>
> Yes, we agree this is a limitation. A more comprehensive method is of course to have such data generated by a high-throughput assay but this is well beyond the scope of a short ML conference paper.
>
>
> >Although overall the manuscript is clear and easy to read, it contains many typos and grammatical errors - see below.
>
> We thank the reviewer for taking the time to point those out. We will be happy to correct all of those in a revised version, if given the opportunity.
>
> >When it comes to in-silico mutations, I am not sure the authors' experiment shows all that much. A better approach might be to see how well the model predicts splicing QTLs as was done in the Pangolin paper (and using integrated gradients works better than mutagenesis in our experience).
>
> The reviewer raises a good point here we should elaborate on. To be clear, this work aims to achieve two objectives: (a) build a model that achieves SOTA performance for tissue specific splicing predictions (b) develop a new BO method for designing RNA sequence with user defined condition specific splicing profile (and formulating the task as such). We are NOT aiming here to prove SOTA in predicting the effect of mutations for say disease diagnostics. We believe such applications are exciting future directions, as we describe in the discussion section. Thus, the in-silico mutation analysis in this work is only aimed to show the model is able to capture tissue specific regulatory elements. Yes, we very much agree overlapping the model predictions with sQTL is a great way to show the ability to predict the effect of mutations but again this is not the focus of this work and many sQTL hits are not tissue specific. We did not have an easy solution/set to assess against (for example, the set of variants used in the recent Wagner et al Nat Gen 2023 is not accessible to the best of our knowledge) and running an extensive analysis for say sQTL that are also tissue specific is beyond the scope of a short ML conference paper. Please also see a much more lengthy discussion of this point, including additional data and analysis we added for another mutation dataset in response to reviewer Bzy6.
>
> We hope the extensive responses we included to all reviewers will convince the reviewer the revised manuscript will safely pass the acceptance criteria.

---

### Official Review · Reviewer_dwAm · 2023-10-31

**Soundness:** 3 good
**Presentation:** 3 good
**Contribution:** 3 good
**Rating:** 6
**Confidence:** 4

**Summary:**

The paper proposes Bayesian optimization approach for designing sequence edits that modify RNA splicing in a desired way. It also proposes a transformer-based model for predicting splicing, which then serves as a surrogate source of truth for the Bayesian optimization method.

**Strengths:**

The main strength of the paper lies with the novel task of sequence editing aimed at altering the splicing outcome.

The proposed technical pipeline is inspired by recent work on Bayesian optimization over structured/discrete domains (cited refs. Maus et al. NeurIPS’22, Stanton et al. ICML’22) and is relatively novel: it combines top-performing current approach for predictive tasks on sequences (Transformers) with latent space Bayesian optimization (LSBO) approach for sequence design.

Experimental results for splicing prediction show that the predictive part of the approach improves upon existing methods. Assessment of the sequence design part resulted in sequences that make biological sense.

**Weaknesses:**

The literature review concerning splice site prediction is phrased in a confusing way. On pg 2, when discussing DNABERT, SpliceAI, etc. authors mention sequence length limit (e.g. 10kbp for SpliceAI) as a key challenge these method face, and yet the method proposed in the paper uses short window, 400bp for each of the four Transformers. Recent tools such as SpliceBERT are not mentioned.

The description of the “condition-specific” nature of the method is somewhat vague, and should be described/discussed in Introduction and in the relevant Methods section in more detail. For example, it seems (pg. 3, bottom paragraph) that the method does not take condition information (RBP knockdown) on input directly, but “simulates it” via sequence modification. On the other hand, “condition” relating to tissue specificity seems to be used as input to the model as part of the Event features; rationale for this approach and its impact on usage of the method e.g. for splicing-altering diseases should be discussed.

**Questions:**

See weaknesses above.

---

> ### Author Response · Authors · 2023-11-22
>
> We thank the reviewer for the time and effort spent to review and comment on our work. We focus below only on the main points raised that require fixing/clarifications.
>
> >The literature review concerning splice site prediction is phrased in a confusing way. On pg 2, when discussing DNABERT, SpliceAI, etc. authors mention sequence length limit (e.g. 10kbp for SpliceAI) as a key challenge these method face, and yet the method proposed in the paper uses short window, 400bp for each of the four Transformers. Recent tools such as SpliceBERT are not mentioned.
>
> We will be sure to improve the literature review, framing the works better in terms of tasks and adding SpliceBERT. Regarding the usage of only short windows - please see a detailed discussion about this point in response to reviewer Bzy6 which raised a similar question. This again is a reflection of lack of clarity in the review section about what each such model/work tries to achieve (which is very different!) and design choices accordingly.
>
> >The description of the “condition-specific” nature of the method is somewhat vague, and should be described/discussed in Introduction and in the relevant Methods section in more detail. For example, it seems (pg. 3, bottom paragraph) that the method does not take condition information (RBP knockdown) on input directly, but “simulates it” via sequence modification. On the other hand, “condition” relating to tissue specificity seems to be used as input to the model as part of the Event features; rationale for this approach and its impact on usage of the method e.g. for splicing-altering diseases should be discussed.
>
> That is a good point which we definitely should make more clear. To address the question raised, the model is “condition specific” in the sense that it learns to predict condition specific splicing outcome. The “condition” can be anything the user decides to train on - tissue type, specific cell line, a specific cell line with an RBP KD (as done for example by ENCODE for hundreds of RBPs), a cellular condition (e.g. stress response in a cell line of interest) etc. This definition is very much inline with previous works on splicing code prediction algorithms, dating back to the first splicing code (Barash et al Nature 2010). In this work we focused on “condition” as “tissue” in two key organisms (human, mouse) as this is a main focus of much current research and relates to therapeutics as well. In this setting we are not trying to mimic an RBP KD as an objective - we only did the in silico motif removal to illustrate that the model is indeed learning condition specific regulatory motifs. Similar analysis showing ability to infer new biology for splicing regulation when modeling RBP KD as a “condition” was shown in Jha et al Bioinformatics 2017 for CELF vs. FOX  antagonistic regulation, which was followed by extensive experimental validation in Gazzara et al Genome Research 2017. We will make sure to improve our description and framing of these tasks in the revision, if given the opportunity.
>
> We hope the extensive responses we included to all reviewers will convince the reviewer the revised manuscript will safely pass the acceptance criteria.

---

### Official Review · Reviewer_Yedq · 2023-10-31

**Soundness:** 3 good
**Presentation:** 2 fair
**Contribution:** 3 good
**Rating:** 5
**Confidence:** 4

**Summary:**

The authors proposes a transformer-based network to predict condition specific splicing event (in terms of psi). The inputs of the network are the RNA sequences near the splicing site, meta data such as sample type (tissue), exon length, intron length, etc. The labels are the psi values quantified using another method called MAJIQ.

The goal of this manuscript is to learn the underlying regulatory mechanisms of splicing using the proposed network such that it could be used as an oracle to predict the splicing based on the sequence information and meta-data. Based on this powerful prediction capability, the authors proposes additional networks to design new RNA sequences to make them splice as expected.

In general, I think the goal is very ambitious. Unfortunately, I am not convinced by the manuscript that the goal is achieved. First, I am not sure what meta data information could be utilised to predict condition-specific splicing, assuming the input RNA sequences are from the same species thus are very similar. I feel it will be useful to give a concrete example to motivate the proposal. Second, I think it will be nice to expand the prediction part with more experiments to validate the basic assumptions. For example, if we replace the input sequences with random sequences or sequences from non-splicing genes, but with the same meta-data, we expect to see that psi is close to 0. Assuming the network has successfully captured the regulatory mechanism, we expect to see the psi will decrease to 0 if we remove the U2 elements (suppl. Fig. 7d) from the input sequences. Since the splicing mechanism is conservative between human and mouse, we expect to see that NN trained on human data should exhibit similar predictive power on mouse homolog genes. If the prediction part works with high precision, then it worth designing novel RNA sequences. It will be really helpful for me to read the ms if the authors could highlight the main ideas and assumptions and hiding unnecessary details.

Minor points:
1. Sec 3.1.1 Fig. 1b => Fig. 1a, second paragraph, "mask the surrounding k tokens", k=?
2. Sec 3.1.2 Fig. 1c => Fig. 1b
3. Explain why you want to center the sequences
4. Right on top of sec. 4, "The Levenshtein constraint is evaluated on ....", why lev(z, z') = lev(\gamma(z), \gamma(z')) hold?

**Strengths:**

The authors proposes a transformer-based network to predict condition specific splicing event (in terms of psi). The inputs of the network are the RNA sequences near the splicing site, meta data such as sample type (tissue), exon length, intron length, etc. The labels are the psi values quantified using another method called MAJIQ.

The goal of this manuscript is to learn the underlying regulatory mechanisms of splicing using the proposed network such that it could be used as an oracle to predict the splicing based on the sequence information and meta-data. Based on this powerful prediction capability, the authors proposes additional networks to design new RNA sequences to make them splice as expected.

In general, I think the goal is very ambitious. Unfortunately, I am not convinced by the manuscript that the goal is achieved. First, I am not sure what meta data information could be utilised to predict condition-specific splicing, assuming the input RNA sequences are from the same species thus are very similar. I feel it will be useful to give a concrete example to motivate the proposal. Second, I think it will be nice to expand the prediction part with more experiments to validate the basic assumptions. For example, if we replace the input sequences with random sequences or sequences from non-splicing genes, but with the same meta-data, we expect to see that psi is close to 0. Assuming the network has successfully captured the regulatory mechanism, we expect to see the psi will decrease to 0 if we remove the U2 elements (suppl. Fig. 7d) from the input sequences. Since the splicing mechanism is conservative between human and mouse, we expect to see that NN trained on human data should exhibit similar predictive power on mouse homolog genes. If the prediction part works with high precision, then it worth designing novel RNA sequences. It will be really helpful for me to read the ms if the authors could highlight the main ideas and assumptions and hiding unnecessary details.

Minor points:
1. Sec 3.1.1 Fig. 1b => Fig. 1a, second paragraph, "mask the surrounding k tokens", k=?
2. Sec 3.1.2 Fig. 1c => Fig. 1b
3. Explain why you want to center the sequences
4. Right on top of sec. 4, "The Levenshtein constraint is evaluated on ....", why lev(z, z') = lev(\gamma(z), \gamma(z')) hold?

**Weaknesses:**

The authors proposes a transformer-based network to predict condition specific splicing event (in terms of psi). The inputs of the network are the RNA sequences near the splicing site, meta data such as sample type (tissue), exon length, intron length, etc. The labels are the psi values quantified using another method called MAJIQ.

The goal of this manuscript is to learn the underlying regulatory mechanisms of splicing using the proposed network such that it could be used as an oracle to predict the splicing based on the sequence information and meta-data. Based on this powerful prediction capability, the authors proposes additional networks to design new RNA sequences to make them splice as expected.

In general, I think the goal is very ambitious. Unfortunately, I am not convinced by the manuscript that the goal is achieved. First, I am not sure what meta data information could be utilised to predict condition-specific splicing, assuming the input RNA sequences are from the same species thus are very similar. I feel it will be useful to give a concrete example to motivate the proposal. Second, I think it will be nice to expand the prediction part with more experiments to validate the basic assumptions. For example, if we replace the input sequences with random sequences or sequences from non-splicing genes, but with the same meta-data, we expect to see that psi is close to 0. Assuming the network has successfully captured the regulatory mechanism, we expect to see the psi will decrease to 0 if we remove the U2 elements (suppl. Fig. 7d) from the input sequences. Since the splicing mechanism is conservative between human and mouse, we expect to see that NN trained on human data should exhibit similar predictive power on mouse homolog genes. If the prediction part works with high precision, then it worth designing novel RNA sequences. It will be really helpful for me to read the ms if the authors could highlight the main ideas and assumptions and hiding unnecessary details.

Minor points:
1. Sec 3.1.1 Fig. 1b => Fig. 1a, second paragraph, "mask the surrounding k tokens", k=?
2. Sec 3.1.2 Fig. 1c => Fig. 1b
3. Explain why you want to center the sequences
4. Right on top of sec. 4, "The Levenshtein constraint is evaluated on ....", why lev(z, z') = lev(\gamma(z), \gamma(z')) hold?

**Questions:**

The authors proposes a transformer-based network to predict condition specific splicing event (in terms of psi). The inputs of the network are the RNA sequences near the splicing site, meta data such as sample type (tissue), exon length, intron length, etc. The labels are the psi values quantified using another method called MAJIQ.

The goal of this manuscript is to learn the underlying regulatory mechanisms of splicing using the proposed network such that it could be used as an oracle to predict the splicing based on the sequence information and meta-data. Based on this powerful prediction capability, the authors proposes additional networks to design new RNA sequences to make them splice as expected.

In general, I think the goal is very ambitious. Unfortunately, I am not convinced by the manuscript that the goal is achieved. First, I am not sure what meta data information could be utilised to predict condition-specific splicing, assuming the input RNA sequences are from the same species thus are very similar. I feel it will be useful to give a concrete example to motivate the proposal. Second, I think it will be nice to expand the prediction part with more experiments to validate the basic assumptions. For example, if we replace the input sequences with random sequences or sequences from non-splicing genes, but with the same meta-data, we expect to see that psi is close to 0. Assuming the network has successfully captured the regulatory mechanism, we expect to see the psi will decrease to 0 if we remove the U2 elements (suppl. Fig. 7d) from the input sequences. Since the splicing mechanism is conservative between human and mouse, we expect to see that NN trained on human data should exhibit similar predictive power on mouse homolog genes. If the prediction part works with high precision, then it worth designing novel RNA sequences. It will be really helpful for me to read the ms if the authors could highlight the main ideas and assumptions and hiding unnecessary details.

Minor points:
1. Sec 3.1.1 Fig. 1b => Fig. 1a, second paragraph, "mask the surrounding k tokens", k=?
2. Sec 3.1.2 Fig. 1c => Fig. 1b
3. Explain why you want to center the sequences
4. Right on top of sec. 4, "The Levenshtein constraint is evaluated on ....", why lev(z, z') = lev(\gamma(z), \gamma(z')) hold?

---

> ### Author Response · Authors · 2023-11-22
>
> We thank the reviewer for the time and effort spent to review and comment on our work. We focus below only on the main points raised that require fixing/clarifications.
>
> >Unfortunately, I am not convinced by the manuscript that the goal is achieved. First, I am not sure what meta data information could be utilized to predict condition-specific splicing, assuming the input RNA sequences are from the same species thus are very similar. I feel it will be useful to give a concrete example to motivate the proposal.
>
> There seems to be some lack of clarity regarding what is “meta data”. As we describe in the text, this data is simply information regarding things like exon length, intron length, tissue type etc. This information is known to relate to splicing outcome (e.g. micro exons that are neuro specific) and are thus useful. We clearly show are useful in the ablation study (Table2). It’s true that in some species splicing may be very different (e.g. in plants) but we are not exploring those cases here.
>
>
> >Second, I think it will be nice to expand the prediction part with more experiments to validate the basic assumptions. For example, if we replace the input sequences with random sequences or sequences from non-splicing genes, but with the same meta-data, we expect to see that psi is close to 0. Assuming the network has successfully captured the regulatory mechanism, we expect to see the psi will decrease to 0 if we remove the U2 elements (suppl. Fig. 7d) from the input sequences. Since the splicing mechanism is conservative between human and mouse, we expect to see that NN trained on human data should exhibit similar predictive power on mouse homolog genes. If the prediction part works with high precision, then it is worth designing novel RNA sequences.
>
> We have worked very hard to add analysis and results that would support our claims that (a) we achieve SOTA performance in tissue specific splicing prediction. We refer the reviewer to the extensive details we included in the response to reviewer Bzy6. Regarding the specific suggestions: Abolition of the splice signal (e.g. U2 binding area) leads to immediate destruction of the splicing - this is part of the pre-training of the Transformer on splice sites which is an easy task shown in many previous papers (and why we didn’t emphasize it). More generally, see our discussion in response to Bzy6 regarding the much easier task of constitutive vs. alternative exons.  We do indeed see similar predictive power between mouse and human as we show in the paper (and see again Bzy6 response).
>
>
> >It will be really helpful for me to read the ms if the authors could highlight the main ideas and assumptions and hiding unnecessary details.
>
> We will make sure to improve the writing and better flush out the main ideas/assumptions, given the opportunity. With that said, we respectfully request the reviewer to carefully consider their rejection score given the above and the many details/clarifications we supplied to the other reviewers. As it stands we did not find any specific claim made by the reviewer that warrants a rejection score. If they still hold the paper should be rejected (score of 5) we would like to have specific arguments about where we fail to develivr on the tasks we set out, namely: (a) develop SOTA predictor for tissue specific splicing (b) use it for a new BO algorithm for designing RNA sequence with tissue specific splicing constraints. We thank the reviewer again for the time and effort on this.

---

> ### Comment · Reviewer_Yedq · 2023-11-22
> **SOTA is not everything**
>
> I think it is equally important to clearly convey the underlying assumptions and show some intuitive examples to the readers beyond SOTA performance, which I have not seen from the authors' feedbacks.

---

> > ### Author Response · Authors · 2023-11-22
> > **Re - SOTA is not everything**
> >
> > Yes! We very much agree it is critical to convey the underlying assumptions and make things clear for readers - both experts and novice. We strived hard to achieve this within the limited format of a conference paper, trying to write both clearly and concisely. Specifically, we would point out that we laid out many of the terms and challenges about splicing, how it is quantified and the challenges involved in its prediction both in the introduction and in a separate supplementary section dedicated to it, including figures. As long time reviewers for all major ML conferences ourselves we try hard to practice what we tell our students and believe the level of explanations we provided far exceeds the standard in those conference papers. We note other reviewers did comment about clarity as a positive thing. With that said, we agree the topic is complex and many terms in the field (being a CompBio task) are sometimes used to mean different things so clarity of assumptions and/or illustrative examples are important. Consequently, in our feedback we also made a very deliberate effort to clarify terms and assumptions - it is simply unfair and untrue to state “have not seen from the authors feedback”. Specific examples of things we addressed in our response are terms used both in the paper and by the reviewer - “Constitutive”, “Alternative”, “in between”, “condition”, and the differences between splicing prediction tasks taken by TrASPr in this work (predicting tissue specific splicing for alternative exons) vs for example SpliceAI (predicting score  for candidate 3’/5’ splice sites and the effect of mutations on those). Bottom line, we tried to address any of the unclear assumptions/terms we identified and included illustrations in the support material. We would be happy to make other clarifications in our response and the paper itself if the reviewer could point to specific issues they believe remain unaddressed:  “clearly convey underlying assumptions” - about what? “intuitive examples” - for what? If specific comments/criticism is given we would be happy to address it.

---

### Meta-Review · Program_Chairs · 2024-01-16

**Metareview:**

Note: This meta-review is written by the PCs

This paper proposes a transformer based approach for predictions of RNA code splicing and also its design. An Bayesian Optimization method is also proposed to produce labelled data. The paper received borderline reviews (6,6,5,5) with two reviewers leaning towards reject. These two reviewers also asked follow up question and do not seem to be entirely convinced. Looking at the conversation, we tend to agree that there are many specific details that remain unclear, especially regarding the experimental comparisons.

Having read the paper myself, I believe the paper is generally difficult to understand for general ICLR audience and improving this aspect may increase the chance of this paper being accepted at a machine learning conference. I was also not able to figure out the exact machine-learning challenge for this problem: we agree that the original problem is difficult but what are the best known ML techniques? Which part of the problem is sufficiently solved and what other parts needs to be addressed? I think a clarity of such big picture will increase not only the chance of it getting accepted but also the impact on the community. I wish the authors all the best for their next submission.

**Justification For Why Not Higher Score:**

Although the paper has many merits, there are many aspects of the paper that can be easily improved.

**Justification For Why Not Lower Score:**

N/A

---

### Decision · Program_Chairs · 2024-01-16

Reject